# The Potential Impact of Selected Bacterial Strains on the Stress Response

**DOI:** 10.3390/healthcare9050494

**Published:** 2021-04-22

**Authors:** Clara Anker-Ladefoged, Thomas Langkamp, Anett Mueller-Alcazar

**Affiliations:** Department of Psychology, Faculty of Human Sciences, Medical School Hamburg, 20457 Hamburg, Germany; clara.anker-ladefoged@studium.fernuni-hagen.de (C.A.-L.); thomas.langkamp@medicalschool-hamburg.de (T.L.)

**Keywords:** microbiome, stress response, probiotics, mental health, stress-induced diseases

## Abstract

Introduction: The composition of the microbiome is subject to a variety of factors, such as eating behavior and the history of medical treatment. The interest in the impact of the microbiome on the stress response is mainly explained by the lack of development of new effective treatments for stress-related diseases. This scoping review aims to present the current state of research regarding the impact of bacterial strains in the gut on the stress response in humans in order to not only highlight these impacts but to also suggest potential intervention options. Methods: We included full-text articles on studies that: (a) were consistent with our research question; and (b) included the variable stress either using biomedical parameters such as cortisol or by examining the subjective stress level. Information from selected studies was synthesized from study designs and the main findings. Results: Seven studies were included, although they were heterogenous. The results of these studies do not allow a general statement about the effects of the selected bacterial strains on the stress response of the subjects and their precise pathways of action. However, one of the works gives evidence that the consumption of probiotics leads to a decrease in blood pressure and others show that stress-induced symptoms (including abdominal pain and headache) in healthy subjects could be reduced. Conclusion: Due to different intake period and composition of the bacterial strains administered to the subjects, the studies presented here can only provide a limited meaningful judgement. As these studies included healthy participants between the ages of 18 and 60 years, a generalization to clinical populations is also not recommended. In order to confirm current effects and implement manipulation of the microbiome as a treatment method for clinical cases, future studies would benefit from examining the effects of the intestinal microbiome on the stress response in a clinical setting.

## 1. Introduction

It has been documented in a large number of studies that stress is a major risk factor for a variety of disorders, such as depression and anxiety disorders, as well as cardiovascular and inflammatory diseases ([1,2,3]). The current stagnation of research into new psychotropic drugs for the treatment of mental disorder [4,5] highlights the urgency of successfully finding new effective options. Recent study results indicate that processes and bacterial culture composition in the microbial ecosystem influence the stress response [6,7,8]. This may signify a new goal-oriented research direction. The growing interest in the human microbiome is most notably reflected in the Human Microbiome Project [9], which aims to identify and characterize the microbiome. The microbiome is one of the most biodiverse ecosystems in the world [10] and its existence and composition are closely related to the development and function of the hypothalamic–pituitary–adrenocortical (HPA) axis [8,11]. The gut provides most of the energy for initiating and maintaining the stress response by reducing its own activity [12]. By limiting the energy available to regulate blood flow and maintain the internal intestinal mucosa, the living conditions for intestinal bacteria (intestinal microbiota) change. This change leads to the death of some and the proliferation of other bacterial cultures, which in turn has effects on the stress response [12,13]. The concept of bacteria with positive effects on mental health, so-called ‘psychobiotics’, was coined by Dinan, Stanton and Cryan in 2013 [14]. Individual studies showed that various probiotics produced similar effects to conventional psychotropic drugs/medications, e.g., for depression and anxiety disorders, but without causing associated side effects [7,14,15]. As early as 1908, Carre, Tissier and Metchnikoff discovered the possibility of replacing harmful bacterial cultures in the intestine by releasing beneficial bacteria and alleviating complaints such as diarrhea in newborns or preventing the outbreak of cholera [15]. Nobel prizewinner Metchnikoff went so far as to hypothesize that probiotics promote host longevity. Manipulation of the intestinal microbiome by administering special bacterial strains as a treatment method for stress-induced diseases is considered a paradigm shift in neuroscience and psychiatry [6]. The high comorbidity of psychiatric symptoms triggered by stress and gastrointestinal disorders, e.g., irritable bowel syndrome or gastric mucosal disorders, highlights the urgency of looking at the gut–brain axis [16]. For instance, half of all patients with irritable bowel syndrome meet the criteria for an affective disorder [17]. Further research in this area, especially in human subjects, is needed to explore potential applications of probiotics and confirm their efficacy for widespread use.

## 2. Theoretical Background

### 2.1. Stress Response

Stress leads to a response in the human body that involves a series of integrated cascades in the nervous, endocrine, and immune systems. Stress in humans triggers two connected pathways: the autonomic, predominantly sympathetic, nervous system (ANS), and the neuroendocrine response which is mediated by the HPA axis. Reciprocal neural connections between the norepinephrinergic neurons of the central nervous system (CNS) and the corticotropin-releasing hormone (CRH) stimulate each other and travel into the systemic circulation.

The ANS is the first main stress pathway, which regulates heart rate (HR), heart rate variability (HRV), breathing and catecholamine release including epinephrine and norepinephrine. The stress response of the ANS provides the most immediate response and can act within seconds [18].

On the contrary, the second main stress pathway is the HPA axis, whose activation leads to the release of glucocorticoids, mainly cortisol in humans, from the adrenal glands. Hormones of the HPA act more slowly (e.g., its peak is usually reached within 10 to 30 min after cessation of an acute psychosocial stressor) due to its hormonal cascade that readies the body for action including the release of physical energy. The HPA axis allows the individual to maintain its homeostasis under acute stress by adapting to increasing demands. Stress-related sensory information is first conveyed to the paraventricular nucleus (PVN) of the hypothalamus which then induces the expression of the hormones CRH and arginine vasopressin, which are then passed toward the pituitary. Here, the adrenocorticotropic hormone (ACTH) is released and passed towards the adrenal glands via the bloodstream where it controls the production and release of cortisol, the major glucocorticoid hormone. Furthermore, cortisol influences various other physiological systems including the CNS, metabolism, cardiovascular function, immune system, muscle tissue, and bones [19]. Cortisol leads to the suppression of the immune response, e.g., the secretion of proinflammatory cytokines. 

In order to protect from overshooting and to keep glucocorticoids within balance, the HPA axis system involves a compound set of interactions and feedback loops between the hypothalamus, the pituitary gland, and the adrenal glands. Although glucocorticoids influence metabolic and immune processes and adapt the organism to changing demands [20], chronic and high stress-induced levels of glucocorticoids can be considered suppressive. Their biological role, however, remains unclear [19]. Under chronic stress, the HPA system is dysregulated which may result in pathophysiological changes and an increased risk for the development of various types of disorders including depression, Cushing syndrome, obesity, diabetes, hypertension, atherosclerosis, osteoporosis, and immune dysfunction [21,22,23]. For example, in animals with intestinal inflammation a prolonged stress response has been shown [24]. In this regard, decreased glucocorticoid receptor expression has been observed in animal models of chronic stress [25].

### 2.2. Intestinal Microbiome and Microbiota

While the intestinal microbiome defines the set of genes of all microorganisms in a given area, microbiota represent the set of microorganisms themselves [26]. The gut is inhabited by 10^13^–10^14^ microorganisms, which is ten times the number of all human cells in the body. Its gene set is 150 times larger than the human gene complement [27]. The estimated number of species colonizing the gut varies in the literature, but it is generally acknowledged that the adult microbiome consists of more than 1000 species and more than 7000 strains [28]. While a healthy intestinal microbiome is characterized by a diverse composition of a wide variety of bacterial species [29], a narrow diversity of bacterial species (dysbiosis) appears to be associated with gastrointestinal diseases and conditions such as allergies, irritable bowel syndrome, or chronic inflammatory diseases such as Crohn’s disease [30]. Bacteria colonize the human intestine perinatally through the exit from the maternal birth canal. Factors such as type of birth (vaginal or cesarean), duration of breast feeding, diet, environment, diseases and their treatment influence the species composition. The number of species grows steadily and stabilizes at the age of two [31], yet remains highly variable across the lifespan [26].

### 2.3. Pro- Pre- and Psychobiotics

Probiotics are live bacteria and any yeasts that are beneficial to human health when administered in adequate amounts [26]. Probiotics are available as dietary supplements or in foods, e.g., yogurt, kimchi, kombucha, pickles, or sauerkraut. The most well-known probiotic bacteria belong to the *Bifidobacterium* and *Lactobacillus* genera. *Bifidobacteria* are part of the naturally existing intestinal microbiota. Bifidobacteria represent less than 10% of the intestinal microbiome [32]. Among other things, they promote the digestive process and have the potential to synthesize certain vitamins (e.g., vitamin B6, folic acid, and thiamine) and reduce toxic metabolites [33]. Just like *Lactobacilli*, they ferment glucose to lactic acid, which stabilizes the acid environment in the intestine to a constant pH (<5). *Lactobacilli* represent about 1% of all bacteria in the intestine. They colonize the small intestine and produce a variety of enzymes that are primarily necessary for breaking down complex carbohydrates, which are afterwards available as nutrition for other bacterial species [34]. Prebiotics, also known as dietary fiber, are indigestible, fermentable carbohydrates that stimulate the growth of any beneficial bacterial group such as *Lactobacilli* and *Bifidobacteria* [15]. Prebiotics can influence the composition of the intestinal microbiota and/or the activity of naturally existing bacteria, and consequently have positive health effects. The combination of prebiotics and probiotics is called synbiotics [31]. Psychobiotics are to be defined as beneficial bacteria (probiotics) or those that support and nourish bacteria (prebiotics) which can influence the connection between bacteria and brain [35].

### 2.4. Microbiome–Gut–Brain Axis

The term microbiome–gut–brain axis refers to the connection between the gut or enteric nervous system (ENS) and the microorganisms living therein and the central nervous system (CNS) [24]. The microbiome–gut–brain axis includes the CNS, ENS, neuroendocrine and immune systems as well as the parasympathetic and sympathetic systems [36]. 

The gut has its own nervous system (the ENS), which is largely autonomous from the brain. Due to its functional activities and interaction with the microorganisms independent of the brain, the term ‘gut brain’ has been introduced. This led to the research field of neurogastroenterology [11]. 

Administration of *Lactobacilli* (*L. casei strain Shirota*) was shown to reduce sympathetic nerve fiber activation by vagus nerve signals in rats [37]. Takada et al. [38] also found increased excitation of the nucleus paraventricularis, which synthesizes CRH and other substances after administering the same species. This suggests that specific species of *Lactobacilli* project signals to the nucleus paraventricularis via visceral neurons as well as through vagus nerve fibers. In a study with mice undergoing vagotomy, no such effects from the administration of probiotics compared to the controls were found [39].

Ninety-five percent of serotonin in the human body is broken down in the intestine, primarily in enterochromaffin cells which produce tissue hormones that serve to control the gastrointestinal tract, as well as in the mucosa and nerve endings of the ENS. Studies with germ-free mice imply that the intestinal microbiome plays a role in tryptophan metabolism which is synthesized to serotonin in the CNS [40]. The strain *Bifidobacterium infantis* 35,624 has been shown to increase L-tryptophan levels, suggesting a role for intestinal microbiota in the synthesis of these indoleamines [41]. The HPA regulates the release of cortisol which, among other things, can influence immune reactivity locally in the gut. O’Mahony et al. [36] suggested that some intestinal microbiota species have the potential to lower inflammatory cytokines and oxidative stress, which may have implications for the stress response as well.

## 3. State of Research

While there is already strong confirmation in the literature that stress can have effects on the gut and intestinal microbiome composition [42,43,44], the intestinal microbiome and the study of bidirectional communication have gained further attention. Most of the studies conducted on the topic were carried out in animals [45,46,47,48,49] due primarily to ethical reasons. Likewise, invasive measures, such as a stool transplant or the ingestion of bacterial cultures, imply the willingness to expose oneself to possible risks and complications. Research design and the results of animal experiments, especially with mice, can only be transferred to humans to a limited extent. Factors such as the type of birth, medical treatment history, social activities, and diet all seem to affect the microbiome [48]. The controlled absence of one or more factors in laboratory mice studies implies that only limited transferability of models to the human microbiome is possible [48].

However, in order to accurately represent the current state of research and the background of human studies, results from animal studies are presented. A widely used study model is the examination of germ-free mice, which grow up isolated from the mother after birth. Bailey et al. [43] demonstrated that the intestinal microbiota is critical to the development of an appropriate stress response of mice later in life. Colonization of the gut must occur within a small-time window in the first few years of life to ensure normal development of the HPA axis [45,49]. In a study using germ-free mice, the increased stress response and cortisol release was restored to normal by administration of probiotics [50]. Other studies with germ-free mice showed abnormalities in brain structures [46,47,49] and increased release of the hormones corticosterone and ACTH [51] compared to a control group. These effects could also be reversed by the administration of probiotics [50]. Differences in stress hormone levels under non-stressed conditions could not be found. The increased stress response in the experimental group could be normalized to basis levels for the most part by means of fecal transplantation from the healthy experimental group. After administering *Bifidobacterium infantis*, they fully regenerated.

In a laboratory study, newborn mice whose mothers were undergoing stress during pregnancy showed dysbiosis of the intestinal microbiota [52]. Stressors that occurred in both early and later life possessed a comparable potential to induce dysbiosis as otherwise seen with antibiotics [43], including an increased intestinal permeability (leaky gut syndrome) after stress onset. This may lead to the transport of multidrug-resistant pathogens through the intestinal mucosa into the bloodstream, among other effects [51]. The increased permeability could be normalized by administration of *Bifidobacteria* (*Bifidobacterium longum 1714*, *Bifidobacterium breve 1205*) [53] and *Lactobacilli* (*Lactobacillus* helveticus, *Lactobacillus rhamnosus*) [54].

## 4. Method

This scoping review was guided by the methodical framework of Arksey and O’Malley [55]. Four databases (PSYNDEX, PsycINFO, PsychARTICLES and PubMed) were systematically searched using keywords derived from the analysis of key studies. Keywords covered the stress variable (cortisol, cortisol awakening response) and the microbiome variable (microbiome, probiotic). An example search strategy including the search terms and hits is shown in Table 1. The full search strategy across all databases is shown in Figure 1.

### 4.1. Identifiying Relevant Studies

#### 4.1.1. Systematic Literature Search

The bibliographic search was conducted on 3rd of December 2020 limited to title and abstract. No differences were detected in using the term gut–brain axis or gut bacteria or microbial ecosystem in comparison to microbiome. As the term microbiome is the most widely used keyword in the relevant literature, it was included in the final search syntax. The terms adrenaline, noradrenaline and heart rate variability did also not provide additional results in comparison to cortisol. The following search syntax was first used in the EBSCO databases PsychARTICLES and PsychINFO: (DE “stress“ OR DE “cortisol“ OR “cortisol awakening response“ OR DE “Hypothalamic Pituitary Adrenal Axis“ OR DE “Hypothalamic Hypophyseal System“) AND (DE “probiotic“) which led to 37 hits, all published by PsychINFO. The same search in PSYNDEX resulted in 17 hits. The process was repeated with a second search syntax: (DE “stress“ OR DE “cortisol“ OR “cortisol awakening response“ OR DE “Hypothalamic Pituitary Adrenal Axis“ OR DE “Hypothalamic Hypophyseal System“) AND (DE “microbiome“), which resulted in 79 hits from PsychINFO and 50 hits from PSYNDEX. To ensure the quality of the search, it was repeated three times with a 2 h interval in all four databases. The search was conducted under consideration of the inclusion and exclusion criteria specified in the following.

#### 4.1.2. Eligibility Criteria

Studies were included if they fulfilled the following criteria: (a) dates of coverage: January 2010–December 2020. The time period of 10 years seems reasonable in order to provide most current data and studies. (b) Original full text was available in German or English; the investigated population was human. The objective of this scoping review is to systematically review scientific evidence on the effects of the intestine microbiome on the human stress response. Therefore, only human studies were included, whereas there was no distinction between clinical and non-clinical trials. (c) The outcome included the collection of biomedical (saliva, urine or blood cortisol levels) or physiological (blood pressure) measures. (d) The usage of a stress questionnaire (e.g., Perceived Stress Scale (PSS) by Cohen, Kamarck and Mermelstein [56] was a possible alternative).

### 4.2. Study Selection

The electronically database search on the 3rd December 2020 yielded 2432 records. After application of database filters, 87 articles remained, which were reduced to 78 after duplicates were removed. The screening of titles and abstracts led to the exclusion of another 65 articles by the author (CAL). Eleven full-text articles were reviewed for inclusion. Seven relevant publications were then identified by means of the pre-specified inclusion and exclusion criteria.

#### Data Extraction

From each study, data were extracted in four categories. The first category, sample characteristics, included information on the sample size, age, gender and exclusion criteria to participate. The second category, outcome measures and stressor, included information on the data sampling method and the specific stressor used in the study. The third category, *interventions*, included the daily administered dose of pre- or probiotics and the duration period of the intake. The fourth category encompassed the main findings of the studies. Data extraction was conducted by CAL, and the extraction forms were verified by AMA accordingly.

## 5. Results

An overview of the included studies is provided in Table 2. The goal of this paper is to elucidate the effects of the intestinal microbiome on the stress response. In order to more comprehensively embed the findings of the studies into this topic, a summary of the results follows. For a more structured presentation, the results were assigned to individual categories (see Section 5.1, Section 5.2 and Section 5.3) to answer the question [57].

### 5.1. Interventions and Impact on the Stress Response

The literature search revealed that all studies that met the inclusion criteria were non-clinical studies investigating the stress response while supplementing probiotics and prebiotics. In these studies, the bacterial cultures administered represent species from the *Lactobacillus* family (*Lactobacillus* casei, *Lactobacillus casei strain Shirota*, *Lactobacillus helveticus* R0052) and *Bifidobacteria* (*Bifidobacterium longum* 1714, *Bifidobacterium longum* R0175). Only one study by Schmidt et al. [63] investigated the effects of two prebiotics (FOS and B-GOS). All studies presented here hypothesized that the microbiota has an impact on the stress response and that this can be positively influenced by ingestion of probiotics and/or prebiotics. This hypothesis was confirmed by five of the seven studies. In these studies, the intake of probiotics resulted in lower cortisol levels as well as reduced psychological stress symptoms compared to the control groups [38,58,59,61,63]. The studies by Möller et al. [62], in which prebiotics were administered, and Kelly et al. [60] did not find any effects on subjective stress levels, salivary cortisol levels or cardiovascular activity. Schmidt et al. [63] found no changes in STAI score or subjective stress level on the PSS by administration of either of the two prebiotics. However, CAR was significantly lower in subjects of the B-GOS experimental group (*p* < 0.05) and they showed lower emotional alertness to negative stimuli in the dot probe task than the control group (*p* > 0.01). In turn, administration of probiotics (*Bifidobacterium longum* 1714) reduced subjects’ daily perceived stress by 15% in the study by Allen et al. [58]. The SECPT, which was used as a stressor, elicited the same response in the subjects, but at a lower stress hormone level overall and without these hormones leading to increased anxiety in the STAI score (*p* < 0.05). The experimental group of Kato-Kataoka et al. [59] had significantly lower salivary cortisol levels than the control group one day prior to testing (*p* < 0.05). Two weeks after testing, a significantly higher fecal serotonin level was also found, although the psychological implications of this result were unclear. 

The duration of intake of pre- and probiotic supplements varies widely. The minimum is 14 days [62] and ranges from three [63] and four weeks [58,59,60,61] to a maximum of eight weeks [38]. Kato-Kataoka et al. [59] additionally surveyed the composition of the intestinal microbiota of his subjects before the start of the intervention, two weeks, one to three days before the test, and two weeks after the test. This was done by analyzing stool samples. They found significantly higher biodiversity of intestinal microbiota species before testing (*p* < 0.05) compared to the control group. This was not observed before the beginning of the intervention. 

That stress affects the composition of the intestinal microbiota has been confirmed in other studies [12,13,43]. From five of the seven studies cited here, it is hypothesized that, in turn, the resulting dysbiosis of the intestinal microbiota may have effects on the stress response. Reduction and mitigation of these negative effects is possibly impacted by taking probiotics. Not only cortisol levels in saliva and urine but also stress-induced physical symptoms, such as stomach aches or headaches, could be reduced. The secondary question of this work, whether pre- and probiotics can influence the human stress reaction, can thus be answered positively.

### 5.2. Stressors and Survey of the Variable Stress

All seven studies presented here collected the variable stress via cortisol levels, either in urine [61] or saliva [58,59,60,63]. Only Möller et al. [62] elicited the variable stress through blood pressure and pulse measurements. In addition to the collection of biomedical parameters, questionnaires were used in six of these studies, with Cohen’s PSS and the STAI being the most used. As anxiety elicits a similar response to stress and stress can also elicit anxiety, studies employing the STAI and the HADS were included in this work [38,59,61,63]. The STAI was used in five of the seven studies [38,58,59,60,63], and Cohen’s PSS was used in four studies [58,60,61,63].

Two of the studies employed natural stressors (academic examination) [38,59], while one study investigated the anticipation of an impending stressful event by using the CAR [63]. Again, three studies used procedures in a laboratory setting [58,60,62] and only one study [61] did not cite an acute stressor. Kelly et al. [60] employed the SECPT, which is a procedure that functions both as a psychological and as a physiological stressor. No significant effects were found in this study.

### 5.3. Sample Characteristics

Subjects varied greatly in terms of sample size and the variables age and gender. The mean number of participants per study was *n* = 63, varying from max. *n* = 140 to a min. *n* = 22. All studies conducted investigations with an experimental and control group. The exact details for the respective group sizes could be found in all publications. In all studies, the control groups were comparable in age distribution. The age range of the subjects was between 18 and 60 years. However, the origin of the subjects and the locations of the survey varied considerably. Two studies each were from Japan [38,59] and Ireland [58,60], while the remainder were from France [61], England [63], and the United States of America [62]. The subjects of all seven studies were healthy volunteers. All studies explicitly excluded the participation of subjects with psychiatric diagnoses, while one of the studies also excluded subjects who had an elevated score on a depression and anxiety scale [61]. The studies by Allen et al. [58] and Kelly et al. [60] also explicitly excluded women as subjects to avoid controlling for the variable menstrual cycle, as this may have effects on cortisol release [64]. This variable, as well as the type of contraceptive method used, were only examined in the study by Schmidt et al. (2015) and were taken into account in the evaluation.

Information on the effect size of the studies can only be found in three of the articles [58,60,61]. With a power of 80%, beta 20% and alpha 5%, these studies met the calculated sample sizes necessary to demonstrate a mean effect of f = 0.3. In doing so, they followed Cohen’s suggestion to set alpha and power to the popular five-eighty convention, i.e., to evaluate the alpha error as four times as bad as the beta error. However, Cohen’s suggestion for this convention is often misunderstood [65], as it is here. The five-eighty-convention may be a good compromise for behavioral research, which Cohen had in mind, if the researcher cannot make a contextual estimate of which of the two errors are worse and the researcher cannot muster resources for appropriately larger samples. It is only for this situation (behavioral research + limited resources + no contextual estimate possible) that Cohen [66] devised the five-twenty convention as a minimum: 

“However, in the judgment of the author, for most behavioral science research (although admitting of many exceptions), power values as large as 0.90–0.99 would demand sample sizes so large as to exceed an investigator’s resources. […] The view offered here is that often, the behavioral scientist will decide that Type I errors, which result in false positive claims, are more serious and therefore to be more stringently guarded against than Type II errors, which result in false negative claims. The notion that failure to find is less serious than finding something that is not there accords with the conventional scientific view. It is proposed here as a convention that, when the investigator has no other basis for setting the desired power value, the value 0.80 be used. […] This 0.80 desired power convention is offered with the hope that it will be ignored whenever an investigator can find a basis in his substantive concerns in his specific research investigation to choose a value ad hoc.”[66], pp. 55–56.

For this clinical research area, the following estimate of the cost of error makes more sense in the eyes of the author (TL):

Consequences of the alpha error: one study declares that probiotic bacteria, on average, have a positive effect on stress, when in fact they do not. Since there are no known side effects so far, the wider use of probiotics in the population is without serious consequences. However, trust in science could be gambled away in the long term if those affected do not perceive any effect or if follow-up studies contradict the original results.

Consequences of the beta error: a study explains that probiotic bacteria on average do not have a positive effect on stress, although they do. Since no side effects are known so far, but stress is a growing problem in some societies, this error can be considered at least as serious. 

Thus, studies in this research area with an alpha error of 0.05 should consider a 0.05 beta too.

### 5.4. Further Results

The results of the included studies in this section represent ancillary results that emerged during the research. In the study by Allen et al. [58], in addition to reducing cortisol levels, taking probiotics led to changes in motility in the prefrontal cortex, which could be detected by an electroencephalogram. Changes during basic learning processes in pair-association learning were shown. The team led by Tillisch et al. [67] found reduced brain activity in sensory and emotional processing areas in response to negative stimuli and processing of emotional cue stimuli (cues) in subjects after four weeks of ingestion of a probiotic milk drink. These findings as well as a reduction in rumination and aggressive thoughts by the intake of probiotics (mixture of two *Bifidobacteria* and five *Lactobacilli* species) could be confirmed by Steenbergen et al. [68].

## 6. Discussion

The number of suitable literature references shows the lack of empirical studies with human subjects and, above all, replication studies. A generalized statement about the effects of the intestinal microbiome on the stress response and the exact pathways is therefore not feasible. The results of the studies presented here suggest that the intake of probiotics and prebiotics, e.g., in the form of dietary supplements, can lead to a reduction in psychological and physiological stress symptoms in healthy adults. This goes along with the findings of the systematic review by Romijn and Rucklidge [69], who then concluded that there is an incomplete evidence base with a need for more research with clinical trials. It should be noted that the review by Romijn and Rucklidge [69] compared subjects with a variety of diseases (including schizophrenia, rheumatoid arthritis) and different habits/characteristics (smokers, regular medication use). The composition of the intestinal microbiome is exposed to a variety of factors such as diet, age, physical activity, place of residence (rural or urban), disease and hygienic standards. Study results from children, adults, healthy subjects vs. subjects with an acute diagnosis or persons of old age from different countries of origin can therefore vary greatly and cannot be generalized. Thus, different results do not necessarily cast doubt on the positive health effects of probiotics on well-being but rather illustrate the limited transferability [15]. McFarland’s [70] systematic review, researching the use of probiotics to correct dysbiosis of normal microbiota, highlights limited transferability. He finds that an acute disease state makes it difficult to determine the baseline composition of the intestinal microbiota and limits the effects of pro- and prebiotics [70].

### 6.1. Interventions and Effects on the Stress Response

Five of the seven studies administered pro- or prebiotics to their subjects in capsule or sachet form. This administration counteracted the death of the live probiotic bacteria before ingestion by the subjects [62]. In contrast, Takada et al. [38] and Kato-Kataoka et al. [59] administered 100 mL of a fermented milk drink to their subjects, each of which was kept below 10° Celsius. A research team from the Technical University Munich found by using different production methods of probiotic supplements (freeze-drying, vacuum drying) that the survivability of bacteria in these processes is dependent on strain affiliation [71,72]. Additionally, according to the final report of the working group “Probiotic Microorganism Cultures in Food” of the Federal Institute for Consumer Health Protection and Veterinary Medicine [73] a certain minimum number of probiotic bacteria is necessary to develop their effects in the human body. According to the report, the recommended daily dose is 10^8^ to 10^9^ per milliliter. This dose was considered by all six studies that administered probiotics to their subjects [38,58,59,60,61,62]. Thus, if the specific bacterial strains had effects on the stress response, the amount administered in the studies was sufficient to detect them. 

The studies cited here examined specific strains of *Bifidobacteria* and *Lactobacilli* of varying concentrations. Therefore, the effects of the intestinal microbiota on the stress response may be limited to specific bacterial strains and cannot be generalized to all species. Changes in stress parameters or behavior could also be due to the direct interaction of the administered bacterial cultures with the microbiota species already present in the gut [74]. The exact modes of action were unclear. This highlights the need for further research defining the specific intestinal microbiota species that have potential effects on the stress response. Establishing and maintaining a high diversity of probiotic species through prebiotics could have more stable and far-reaching positive effects. Simply giving probiotics may only produce very short-term effects, and the presence of prebiotics might be of help. The recommenced increase in the subjective stress level in the two-week follow-up period of the subjects who previously had a low level due to the intake of probiotics underlines this [58]. 

The duration of ingestion of the bacterial cultures varies among the studies. An intake period of 14 days [62] might be too short to achieve measurable effects, if any. Studies that found effects administered probiotics to their subjects for at least three weeks [63], predominantly four weeks [58,61], or eight weeks [38,59]. Only one of the studies examined the baseline intestinal microbiota of the subjects before the study and analyzed it during and after the intervention [59]. The healthy subjects in the studies that did not find any effects may have already had an intestinal microbiota composition that was healthy for them [60,62]. Additionally, the administered dose might have been too low.

Results from studies examining the effects of exam stress on eating habits suggest that stress leads to higher consumption levels and more frequent consumption of sweets and fast food [75,76,77]. Eating habits of the subjects of Takada et al. [38] and Kato-Kataoka et al. [59] were not further documented. Therefore, it is possible that the results of these studies are subject to additional factors which influence the diet. It is conceivable that inappropriate food choices may result in an inadequate supply of intestinal microbiota and thus may have stress-aggravating effects. One study demonstrated that 80% of the subjects studied (*n* = 218) maintained a healthy diet under normal circumstances, but only 30% (*n* = 82) did so under stress [76]. 

The studies by Möller et al. [62] and Kelly et al. [60] emphasize the limitations of transferring results from studies with animals to humans. In contrast to Bravo et al. [39], who obtained lower levels of corticosterone and depressive behaviors in mice compared to a control group by administering the same species (*Lactobacillus rhamnosus*) over a 28-day period, Kelly et al. [60] were not able to translate this effect to humans. Schmidt et al. [63] found reduced reactivity to negative stimuli after administration of probiotics. Thus, administration of these probiotics could influence the perception and appraisal of the threat and stressor. The results of these studies provide counter-intuitive evidence that the behavioral effects of probiotics in animals [39] can be extended to the processing of affective stimuli in humans. Further research and replication of these studies is needed. The supplementary results of the study by Schmidt et al. [63], showing that the control and FOS experimental group with elevated cortisol levels responded more quickly to negative stimuli and reacted more quickly to negative stimuli than to neutrals, suggest that higher stress levels are associated with increased attention to negative stimuli. The faster reaction time of the B-GOS experimental group to the neutral stimuli are in agreement with the results of studies with subjects administered citalopram, an SSRI, or the benzodiazepine diazepam [78,79].

In Kato-Kataoka et al. [59], analysis of the composition of the intestinal microbiome using genetic analysis indicated that the intake of probiotics led to an increase in the diversity of intestinal microbiota species. Another study with depressive patients showed that the affected individuals had a low diversity and a higher level of *Bacteroidaceae* than healthy subjects [80]. The number of *Bacteroidaceae* in the control group of Kato-Kataoka et al. [59] was significantly increased one day before the school examination, and the intake of probiotics was able to reduce it. Thus, the administration of probiotics could complement and improve the treatment of depression.

In addition to the promising results of the studies cited here, there is also evidence of the potential side effects of probiotics. Rao et al. [81] studied subjects who suffered from severe stomach problems (bloating, stomach pain) and concentration problems that occurred shortly after eating. They showed that these subjects had a greatly increased number of *Lactobacilli* in the small intestine, which produce lactic acid. In normal colonization, only small amounts of lactic acid are produced there, posing no danger to the body. Lactic acid can pass through the intestinal wall into the bloodstream, which has a toxic effect on neurons in the brain when transported in large quantities [81]. This study was not included in this work due to the exclusion criteria.

### 6.2. Stressors and the Survey of the Variable Stress

Schmidt et al. [63] asked their subjects to independently collect saliva samples immediately upon waking and then every 15 min until reaching one hour. When exposed to chronic stress, morning cortisol levels in saliva rise above the normal limit (between 7:00 and 9:00 a.m., 0.6–8.4 μg/L) [82]. Since this was the ‘normal’ wake-up time of the subjects, it can be assumed that the daily rhythm adapted to this within certain limits, meaning that the results retain their validity [83]. The validity of the CAR measurement depends to a large extent on compliance with the timing of the sample collection [84]. As this was done independently by the test subjects without supervision, the validity may have been reduced as a result. However, the ecological validity is given by the everyday setting in the own household. The results of the studies, which used natural stressors such as an exam in studies, are not transferable to clinical populations (e.g., depression, anxiety disorders).

Only the study by Möller et al. [62] elicited the variable stress via cardiovascular activity, using blood pressure and pulse measurements. In another study, which followed subjects over 16 years, it could be shown that an increased cardiovascular stress response increased cardiovascular morbidity and mortality [85]. An additional questionnaire, which queried the assessment of the stress procedure in this study (PASAT), showed a reduction in the perceived difficulty and stress of the procedure as well as a large reduction in cardiovascular activity at the second examination. This indicates an adaptation to the procedure which led to a weaker stress response. No effects of probiotics on the stress response were found here, which may be due to an insufficiently strong stressor. This also suggests that the effects of probiotics may only be demonstrated in the presence of severe stressors.

### 6.3. Sample Characteristics

The transferability of the results represented by clinical populations is questionable. The studies cited here could not explicitly support research into the use of these bacterial cultures for the treatment of psychiatric disorders due to healthy subject groups. It is possible that the effects of specific microbiota strains on the stress response may only become evident in the presence of more severe symptomatology or greater variability. Thus, the studies cited here also offer important results as to whether the effects unfold across the spectrum of a trait or whether they are only effective under extreme conditions. Further research in the clinical setting is needed to explore and confirm its potential use as a treatment modality. Moreover, all studies conducted research with adults between the age of 18 and 60, which limits their generalizability to children as well as to the elderly. A lack of colonization of the gut by microbiota in the first years of life could lead to a dysfunctional development of the HPA, which might increase vulnerability to mental disorder [86]. This complicates transferability to this specific group and requires consideration in future sub studies.

Countries of origin varied widely. Student subjects from Japan might be accustomed to higher levels of stress than subjects from France, for example, due to the performance culture prevalent there [87]. According to Lazarus cited in Biggs et al. [88], the perception of stressors and the evaluation as such is individual and dependent on previous experiences and the available coping strategies. Similarly, eating habits differ greatly. Food, customs, and even hygiene differ in many countries and cultures, which means that the composition of the intestinal microbiota can vary greatly.

Here, it is important to test the findings of animal studies [45,49] on behalf of their validity in humans. Additionally, altered composition of the intestinal microbiome and reduction in *Bifidobacteria* and *Lactobacilli* in the population aged 60 years and older [89] complicates transferability to this specific group and requires consideration in future sub studies.

## 7. Limitations

The aim of this scoping review was to provide an exhaustive survey of the state of research on the impact of the intestinal microbiome on the stress response. The included studies offer an orientation about the subject of investigation and show perspectives about possible developments. The cited studies all administered different specific strains of pre- and probiotics to their specific group of participants, which would indicate that the effects could have been random finds. There is a limitation in that the selection and content processing of the literature was only done by one person. This leaves the results subject to a certain subjectivity and potential errors. This refers to the selection of the databases, the formulation of the inclusion and exclusion criteria, the exclusion of articles requiring payment, and the interpretation of the results. It should also be noted that the results of this work cannot be generalized. The number of thematically processed studies is very small, which limits a complex and differentiated presentation. 

## 8. Conclusion and Future Directions

The included studies provide a perspective on the possible positive mechanisms of action of the intestinal microbiome on the stress response. Chronic stress can alter the composition and balance of the intestinal microbiome, which can lead to a variety of diseases. Daily intake of probiotics could mitigate this negative effect and maintain the balance of microbiota species in the gut. The improvement in stress-induced symptoms (including abdominal pain and sleep problems) as well as the reduction in cortisol levels in blood and urine in healthy subjects advocate the positive effects of probiotics on well-being. While psychoactive drugs, such as diazepam, pose risks such as addiction and memory problems [7,14], none of these effects of probiotics and prebiotics on subjects were found in the included studies. In the future, studies investigating mental disorders related to stress should include the intestinal microbiome as an important regulator of the HPA so that specific interventions can be developed to target the microbiome. Long-term studies with clinical groups are necessary to investigate the role of the intestinal microbiome on the stress response and stress-related diseases in order to prevent or treat them.

## Figures and Tables

**Figure 1 healthcare-09-00494-f001:**
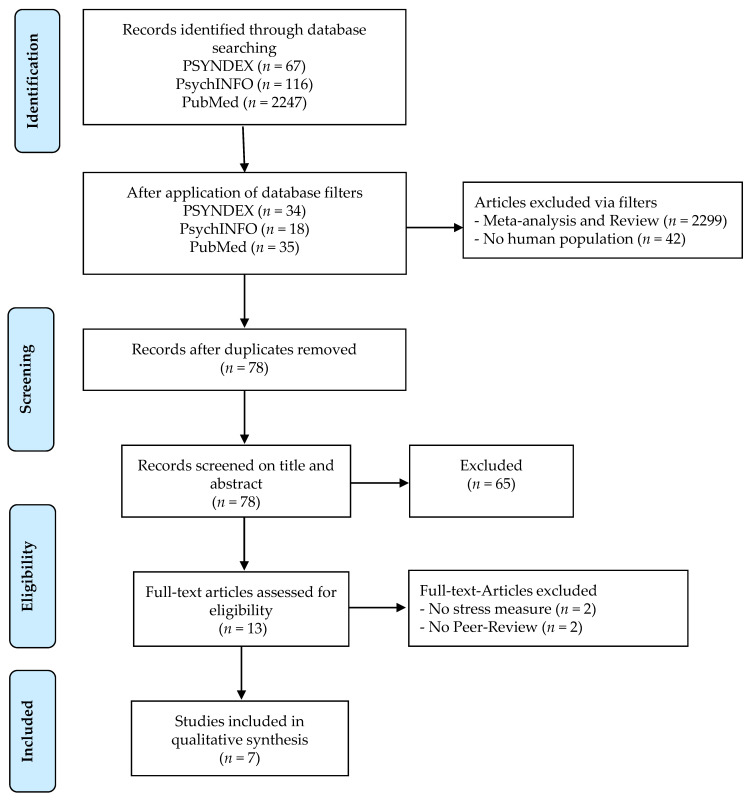
Flow chart of literature search and study selection. *n*, number.

**Table 1 healthcare-09-00494-t001:** Search strategy for all three databases using the same search terms/combinations.

Database and Date of Search	Hits	Exclusion of Meta-Analysis and Reviews	Additional Filters
PsychINFO 3 December 2020	*n* = 116	*n* = 56	Dates of coverage 2010–2020: Species human: *n* = 18
PSYNDEX 3 December 2020	*n* = 67	*n* = 35	Dates of coverage 2010–2020: *n* = 34 Species human: n = 34
PubMed 3 December 2020	*n* = 2.247	*n* = 40	Dates of coverage 2010–2020: *n* = 39 Species human: *n* = 35

**Table 2 healthcare-09-00494-t002:** An overview of the included studies.

First Author and Year	Sample Characteristics	Outcome Measures and Stressor	Intervention	Main Findings and Statistical Limitations
Allen et al. [58]	Planned sample size of one-way ANOVA: 20 (alpha/beta/eta: 0.05/0.20/0.3) Actual sample size: 22 Age, mean (SD) years: 25.5 (1.2) Gender, *n* (%) male: 22 (100) Exclusion criteria: women, chronic diseases, psychiatric diagnosis, following a diet, regular intake of medication Country: Ireland	Primary: Salivary cortisol Secondary: State-Trait Anxiety Inventory (STAI), PSS Stressor: Socially Evaluated Cold Pressure Test (SECPT)	Daily Intake of: one probiotics stick containing *Bifidobacterium longum* 1714 or placebo Duration: four weeks and a two-week follow-up period without intake of probiotics or placebo	Daily perceived stress of the experimental group after four weeks was significantly lower by 15% compared to control group (*p* < 0.05). Daily perceived stress increased again in the follow-up period. SECPT triggered an equal stress reaction at an overall lower stress hormone level and without leading to an increased subjective anxiety on the STAI score (*p* < 0.05). Limitations: Unreasonable beta and eta. No extra power analysis for non-parametric tests => Probably all tests underpowered.
Kato-Kataoka et al. [59]	Planned sample Size: unknown Actual sample size: 47 Age, mean (SD) years: 22.8 (0.4) Gender, *n* (%) female: 21 (45); male: 26 (55) Exclusion criteria: age > 30 years, taking medication three months prior to enrollment Country: Japan	Primary: Salivary Cortisol Secondary: STAI, stool samples, Visual Analogue Scale (VAS), Gastrointestinal Symptom rating scale (Japanese version); GSRS) Stressor: School examination	Daily intake of: 100 mL of milk fermented with *L. casei strain Shirota* or placebo Duration: 56 days	Subjects in the experimental group had significantly lower salivary cortisol levels than the control group one day before their examination (*p* < 0.05). No effect on STAI scores but significantly lower expression of the subjective feeling of stress on the VAS. Reduction in stress-induced physical pain and cold symptoms (*p* < 0.05). Limitations: Kato-Kataoka et al. [59]: “The major limitation of this study was its lack of statistical power because of its small sample of participants.”
Kelly et al. [60]	Planned sample size of one-way ANOVA: 20 (alpha/beta/eta: 0.05/0.20/0.3) Actual sample size: 29 Age, mean (SD) years: 24.69 (0.75) Gender, *n* (%) male: 29 (100) Exclusion criteria: chronic diseases, regular intake of medicine or antibiotics, following a diet Country: Ireland	Primary: Salivary Cortisol Secondary: Blood sample for cytokine measurements, PSS, STAI Stressor: SECPT	Daily intake of: one capsule containing probiotics (*Lactobacillus rhamnosus*) or placebo Duration: four weeks, then conditions for experimental and control group were switched	No effect of probiotics on subjective stress levels (before and after SECPT), salivary cortisol levels in response to the SECPT, PSS and STAI scores, compared with control group. Increase in cytokine levels in the placebo phase which was not significant. Limitations: same as Allen et al. [58]
Messaoudi et al. [61]	Planned sample size of U-Test: 56 (alpha/beta/effect size: 0.05/0.20/unknown) Actual s Sample size: 55 Age, mean (SD) years: 42.8 (8) Gender, *n* (%) female: 41 (75); male: 14 (25) Exclusion criteria: psychiatric, neurological or cardiovascular disease, allergies, regular intake of vitamins Country: France	Primary: urinary cortisol over a 24-h period Secondary: Hospital and Depression Anxiety Scale (HADS), Coping Checklist, PSS Stressor: no acute stressor, normal everyday life of subjects	Daily intake of: one probiotic stick containing 1.5 g of two species (*L. helveticus* R0052, *B. longum* R0175) Duration: 30 days	Lower urinary cortisol levels and subjectively lower stress level in the experimental group (*p* < 0.05). Lower depression scale subscore on the HADS (*p* < 0.01) and higher positive re-evaluation (*p* < 0.05), lower self-blame (*p* < 0.05), higher problem-solving competence in the Coping Checklist (*p* < 0.05). No Effects were found for both groups on the results of the PSS. Limitations: Unreasonable beta and no clear statement on the assumed effect size. G*Power gives a planned sample size of 60 if we assume a large effect for a two-tailed U-Test => Tests probably underpowered since found effects are smaller.
Möller et al. [62]	Planned sample size: none Actual sample size: 105 Age, mean (SD) years: 20.17 (1.26) Gender, *n* (%) female: 69 (66); male: 36 (34) Exclusion criteria: bowel disease, taking antibiotics three months prior to enrollment, regular intake of pro- or prebiotics	Primary: Blood pressure, pulse Secondary: stress procedure evaluation Stressor: Paced Auditory Serial Addition Test (PASAT)	Daily intake of: one probiotic capsule containing three *Bifidobacterium* and five *Lactobacillus* species Duration: 14 days	No significant effect of probiotics intake on blood pressure or pulse rate (*p* < 0.05). Limitations: Explorative study with no post hoc power analysis. However, a power of 0.79 is given if we use *n*/alpha/eta as 105/0.05/0.1 for an ANOVA with repeated measures, within-between interaction and otherwise G*Power default values.
Schmidt et al. [63]	Planned sample size: none Actual sample size: 45 Age, mean (SD) years: 23.27 (3.89) Gender, *n* (%) female: 23 (51); male: 22 (49) Exclusion criteria: DSM-IV diagnosis, gastroenteric, neurological or immune disease, taking antibiotics three months prior to enrollment, regular intake of pre- or probiotics	Primary: Salivary Cortisol Awakening Response (CAR) Secondary: STAI, PSS, Dot probe-task Stressor: no acute stressor, anticipation of a stressful event of the upcoming day	Daily intake of: one of two probiotic sticks (fructooligsaccharides (FOS) or Bimuno™-galacto-oligosaccharides (B-GOS)) or placebo Duration three weeks	CAR in saliva was significantly lower in the B-GOS experimental group compared to the control group (*p* < 0.05). The experimental group also showed significantly lower alertness to emotional stimuli (*p* > 0.01) in the dot probe task. No effects could be found for the FOS group and no changes in STAI or PSS scores were shown in both experimental groups. Limitations: No sample size planning. The study claims that underpowered results were not interpreted, but does not provide any information on how large the minimum power was set or calculated => Power unknown.
Takada et al. [38]	Planned sample size: none Actual sample size: 140 Age, mean (SD) years: 22.9 (0.2) Gender, *n* (%) female: 64 (46); male 76 (54) Exclusion criteria: subjects > 30 years, diagnosis of mental disorder and a score of ≤60 on the Self-rating Depression scale, regular intake of pre- or probiotics	Primary: Salivary cortisol levels Secondary: STAI Stressor: School examination	Daily intake of: 100 mL milk fermented with *L. casei strain Shirota YIT* 9029 or placebo Duration: eight weeks	Significant reduction in cold symptoms (sore throat, headache, fever) at week 5–6 (*p* < 0.05) and abdominal region pain at week 7–8 (*p* < 0.05) in the experimental group. The group as well had significantly lower cortisol level (*p* < 0.05) one day before the exam. Significant changes were found for STAI scores in comparison to the control group (*p* < 0.01). Limitations: No sample size planning. Power unknown.

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
