# Peer review of "The Potential Impact of Selected Bacterial Strains on the Stress Response"

_healthcare, 2021, doi:10.3390/healthcare9050494_

Round 1
Reviewer 1 Report
The manuscript summarizes current studies to elucidate the effects of the intestinal microbiome on the stress response and is well written. I believe that this manuscript is suitable for publication in Healthcare.
Minor comments;
In line 37, ‘(‘ is correct?
In line 118, ‘Bacterial colonize …’ is correct? ‘Bacteria colonize …’?
In line 123, what means ‘andy’ ?
In line 129, ‘Bacteriodes’ is correct? Bacteroides?
In line 278-280, is ‘microbiome’ suitable for this sentence? Because the authors define ‘microbiome’ as the set of genes of all microorganisms in line 108, does this sentence mean that bacterial genome (not living bacteria) has an impact on the stress response? Please write carefully.
In the whole sentence, the names of bacterial genera and species should be written in italics. Please correct them. For examples, ‘Bifidobacterium’, ‘Lactobacillus’, ‘L. casei’, etc should be italic.
In table 1, in the row of PsychINFO, in the column of additional filters, the number of ‘Dates of coverage 2010-2020’ lacks. I think.
In table 1, I wonder that the number of ‘Dates of coverage 2010-2020’ (n=39) is too small in comparison with the number of ‘Hits’ (n=2247). Please confirm again.
Author Response
Please see the attachment.
We thank you and the reviewers for this evaluation of our manuscript as well as providing us with such insightful comments. We are happy to include the reviewer comments. We believe that those changes have significantly improved the manuscript and we look forward to your decision. Please find a point-by-point response to each concern or problem below. The revision has been approved by all co-authors. Additions to the manuscript have been highlighted using the review markups, deletions are marked via crossing-out in the actual manuscript. The clean version is also attached for better readability.

Reviewer 2 Report
Anker-Ladefoged et al article “The impact of the intestinal microbiome on the stress response” is a scoping review on the effects of different probiotics on stress and stress-related diseases, although the report actually focuses on seven articles. These works were selected by a flow chart of literature search that also includes a step based on subjective eligibility. The seven articles are deeply studied and discussed. However, in order to answer the objectives established by the authors and evaluate the global interest of the report, there are certain important general questions that need to get addressed.
First of all, it does not seem totally clear or explicit the way the authors reached a selection of only seven articles starting from 2432 records. It seems unlikely that only seven articles are relevant in such a wide field as this. Maybe this could be more explicitly explained in section 4.2, where they only indicate that one author excluded most of the reviewed papers just by screening title and abstract. They themselves admit this weakness in the Limitations section of the paper (section 7, lines 546-549).
Second, the seven articles are about different probiotic strains, different population groups and make different claims. Therefore, it is difficult to get general conclusions by comparing them. The FAO/WHO definition of probiotics as well as several consensus statements of the International Scientific Association for Probiotics and Prebiotics released in the last few years underline the idea that specific claims must be restricted to specific bacterial strains and target populations.
Therefore, since it is impossible to achieve the objective announced in the title, maybe the title itself, the abstract and the scope of the paper could be restructured and redefined in order to clearly indicate the reader what he/she should expect from the beginning.
Other than that, I have several indications and/or typos that I need to be corrected in the manuscript:
Lines 23-25. Abstracts should be self-sufficient since some people may not have the possibility of accessing the references. Therefore, I would avoid including [1] and [2]. Instead, authors can just say: “One of the works gives evidence that the consumption”…(…)… “and other shows that stress-induced…” since these are the object of the work itself, rather than references needed for discussion.
Line 37. “(including [3-5]”. Please close the round brackets or remove the initial one.
Line 111. “The gut is inhabited by 1013-1014 microorganisms, which is ten times the number of all human cells in the body.” Please, give a reference for this sentence. The ten times estimate was given by Savage in 1977 and many authors have discussed it lately. It is not essential data but, if mentioned, it is sufficiently important as to give a current reference.
Line 114. “strands”. I think authors mean “strains”. They call strands to bacterial strains throughout the manuscript. See also lines 426, 429, 231 and 520.
Line 123. “andy”. Remove “y”.
Line 127. “belong to the Bifidobacterium and Lactobacillus family”. Bifidobacterium and Lactobacillus are not families but genera and they do not belong to the same family. In fact, they are very different bacterial taxa, belonging to even different phyla, the deepest taxonomical division of the Bacteria domain. Also, italics are used for bacterial and viral taxa at the level of family and below (but not in the specific denominations of strains and serovars). The use of italics should be kept throughout the manuscript as well.
Lines 128-129. “Together with the bacterial species Bacteriodes and Eubacteria” Bacteriodes does not exist. I guess it is Bacteroides, and it is not a species but a genus. Eubacteria is not a species either. It is an old fashion name to design the domain Bacteria that is no longer used nor recommended.
Line 129. “they represent 75% of all microorganisms existing in the intestine”. For the reasons indicated above, this is not true. I think authors might mean that Bacteroidetes and Firmicutes (two bacterial phyla) represent the 75% of all microorganisms (although I think the percentage is higher and the one of Bifidobacteria by itself is far much less, actually). Please, check and quote with a reference to support this claim.
Lines 135-136. “Afterwards, they are available as nutrition for other bacterial species”. "They" refers to “carbohydrates”, does not it? Maybe it would be better to rephrase it because it took me to read the sentence twice or three times to understand since I thought "they" referred to “lactobacilli”. Maybe: “for breaking down complex carbohydrates, which are afterwards available as nutrition for other bacterial species”.
Lines 136-138. “Prebiotics, also known as dietary fiber, are indigestible, fermentable carbohydrates that stimulate the growth of lactobacilli and bifidobacterial” This sentence gives the impression that only these two bacterial groups are stimulated by prebiotics, and in fact, prebiotics are defined as to stimulate any beneficial bacterial group. Also “bifidobacterial” is an adjective. Maybe authors mean bifidobacteria (as the layman plural of Bifidobacterium) or “Bifidobacteriales” which is a taxonomical group at the level of order.
Line 149. “(O'Mahony et al., 2011)” Please, keep using the numeral system for references.
Line 153. “L. Casei” is a scientific name, it has to be in italics and the species part in lowercase (L. casei). I know the name is miswritten in the actual reference but still, the correct typing should be uncapitalized.
Lines 164-165. “The bacterium Bifidobacterium infantis 35624” A bacterium is the way to refer to a single cell. Instead, B. infantis 35624 is a strain and it would be more accurately named as “The strain Bifidobacterium infantis 35624”.
Line 168. “O'Mahony et al. [27]”. Reference 27 is Cryan and O’Mahony. Authors may mean ref. 43 (O’Mahony et al). Also, please check the title of reference 27 because it seems to be duplicated.
Line 186. “und” please change to and. Also, reference 41 is Bailey et al. I think authors may mean reference 44 (Bailey and Coe).
Line 197. “Bifidobacterium infantis” in italics
Line 204-206. Italics in (Bifidobacterium longum 1714, Bifidobacterium breve 1205) [52] and lactobacilli (Lactobacillus helveticus, Lactobacillus rhamnosus).
Fig. 1. “Meta-analysis and Reviw” Please, include the “e”
Line 248. Please, erase the line in blank.
Lines 249-254. The figures indicated here do no coincide with the ones indicated in Fig. 1. Please, check.
Line 252. Is it unusual to indicate which author does what. The same on lines 262-263. Usually this goes at the end of the article.
Line 255. The font of this section title is not concordant with the other sections.
Table 2. “three species of bifidobacterias and five species of lactobacillus” plural of bacterium is bacteria (without s) and the genus must be in capital italic letters. Actually, I think authors mean “three Bifidobacterium and five Lactobacillus species”
Also in the table, Bimuno- is a commercial name. Does it need the trademark symbol?
Line 275. “Lactobacillus Shirota” is not an approved name. It is Lactobacillus casei Shirota, as already indicated earlier.
Line 278. “prebiotic cultures”. Prebiotics are not living beings and therefore they are not cultures.
Line 284. “Kelly et al. (2017)” Please indicate the number of the reference. Is the [1] at the end of this sentence appropriate?
Lines 306-308. “From five of the studies seven cited here, it is hypothesized that, in turn, the resulting dysbiosis of the intestinal microbiota may have effects on the stress response.” As indicated earlier, each one of these articles may have their own conclusions but it seems difficult to withdraw a global conclusion from all of them, even more considering that only some of them actually report effects. Also, I guess it is “the seven studies cited here”.
Line 330. “N=63 variing from max. N=140 to a min. N=22”. In the rest of the manuscript “n” is used instead of “N”. Also, "varying" instead of “varying”.
Line 425. “108 to 109”. Please, correct 108 to 109
Lines 437-439. “Simply giving probiotics, which die off quickly without the presence of prebiotics, may only produce very short-term effects.” This sentence seems extremely simplistic and not accurate at all. There are a high and mostly not evaluated number of variables affecting the survival of a bacterial strain throughout the gastrointestinal tract and I do not think any strain survival is dependent on the presence of prebiotics. I would rephrase in something like “Simply giving probiotics may only produce very short-term effects, and the presence of prebiotics might be of help”
Lines 459-463. Please include the numbers in the four references named in these lines.
Author Response

(The authors gave the same response as above.)

Round 2
Reviewer 2 Report
Dear authors, thank you very much for addressing my comments and making the appropriate changes. I only need to pinpoint four of them that still need some minor correction.
- Lines 128-129: (former comment 8) “belong to the Bifidobacterium and Lactobacillus family”. Bifidobacterium and Lactobacillus are not families but genera and they do not even belong to the same family. This has not been corrected. It could be changed to “belong to the Bifidobacterium and Lactobacillus genera”.
- Lines 168-169: (former comment 16) Please, check again the reference number. I think in the new version it is 36 instead of 37 (O’Mahony et al., 2011).
- Line 187: (former comment 17: “reference 41 is Bailey et al. I think authors may mean reference 44 (Bailey and Coe)”). Please, check the reference number again. In the new manuscript, it is 45.
- Fig. 1 (former comment 20). “Meta-analysis and Reviw” Please, include the “e”. This remains uncorrected in the figure (line 239).
Thanks again and best wishes on your work.
Author Response
Response to comments on healthcare-1142096 – Minor Revisions
Dear Sonia Yang, dear colleague,
We thank you and the reviewer for this evaluation of our manuscript as well as providing us with some more insightful comments. We are happy to include the reviewer comments. Please find a point-by-point response to each of the four concerns below. The revision has been approved by all co-authors.
Reviewer 2:
Comment: Lines 128-129: (former comment 8) “belong to the Bifidobacterium and Lactobacillus family”. Bifidobacterium and Lactobacillus are not families but genera and they do not even belong to the same family. This has not been corrected. It could be changed to “belong to the Bifidobacterium and Lactobacillus genera”.
Response: We thank the reviewer for this remark and adjusted the sentence according to the suggestion.
Comment: Lines 168-169: (former comment 16) Please, check again the reference number. I think in the new version it is 36 instead of 37 (O’Mahony et al., 2011).
Response: We also corrected the reference number above.
Comment: Line 187: (former comment 17: “reference 41 is Bailey et al. I think authors may mean reference 44 (Bailey and Coe)”). Please, check the reference number again. In the new manuscript, it is 45.
Response: Here, the correct reference should be Bailey et al (2011) as in the former version. Thanks for clarifying. We changed it accordingly.
Comment: Fig. 1 (former comment 20). “Meta-analysis and Reviw” Please, include the “e”. This remains uncorrected in the figure (line 239).
Response: We acknowledge the reviewer’s remark and altered the sentence according to the suggestion.